# Decentralized Attribution of Generative Models

**Changhoon Kim[1],*, Yi Ren[2],*, Yezhou Yang[1]**
School of Computing, Informatics, and Decision Systems Engineering[1]
School for Engineering of Matter, Transport, and Energy[2]
Arizona State University
{kch, yiren, yz.yang}@asu.edu

## Abstract

Growing applications of generative models have led to new threats such as malicious personation and digital copyright infringement. One solution to these threats is model attribution, i.e., the identification of user-end models where the contents under question are generated from. Existing studies showed empirical feasibility of attribution through a centralized classifier trained on all user-end models. However, this approach is not scalable in reality as the number of models ever grows. Neither does it provide an attributability guarantee. To this end, this paper studies decentralized attribution, which relies on binary classifiers associated with each user-end model. Each binary classifier is parameterized by a user-specific key and distinguishes its associated model distribution from the authentic data distribution. We develop sufficient conditions of the keys that guarantee an attributability lower bound. Our method is validated on MNIST, CelebA, and FFHQ datasets. We also examine the trade-off between generation quality and robustness of attribution against adversarial post-processes.[1]

## 1 Introduction

Recent advances in generative models (Goodfellow et al., 2014) have enabled the creation of synthetic contents that are indistinguishable even by naked eyes (Pathak et al., 2016; Zhu et al., 2017; Zhang et al., 2017; Karras et al., 2017; Wang et al., 2018; Brock et al., 2018; Miyato et al., 2018; Choi et al., 2018; Karras et al., 2019a;b; Choi et al., 2019). Such successes raised serious concerns regarding emerging threats due to the applications of generative models (Kelly, 2019; Breland, 2019). This paper is concerned about two particular types of threats, namely, malicious personation (Satter, 2019) , and digital copyright infringement. In the former, the attacker uses generative models to create and disseminate inappropriate or illegal contents; in the latter, the attacker steals the ownership of a copyrighted content (e.g., an art piece created through the assistance of a generative model) by making modifications to it.

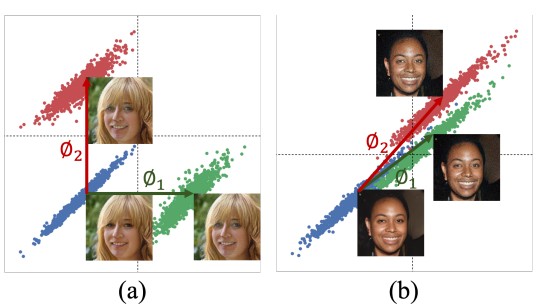

Figure 1: FFHQ dataset projected to the space spanned by two keys $\phi_1$ and $\phi_2$. We develop sufficient conditions for model attribution: Perturbing the authentic dataset along different keys with mutual angles larger than a data-dependent threshold guarantees attributability of the perturbed distributions. (a) A threshold of $90 \deg$ suffices for benchmark datasets (MNIST, CelebA, FFHQ). (b) Smaller angles may not guarantee attributability.

We study model attribution, a solution that may address both threats. Model attribution is defined as the identification of user-end models where the contents under question are generated from. Existing

---

*Equal contribution.

[1]https://github.com/ASU-Active-Perception-Group/decentralized_attribution_of_generative_models

studies demonstrated empirical feasibility of attribution through a centralized classifier trained on all existing user-end models (Yu et al., 2018). However, this approach is not scalable in reality where the number of models ever grows. Neither does it provide an attributability guarantee. To this end, we propose in this paper a decentralized attribution scheme: Instead of a centralized classifier, we use a set of binary linear classifiers associated with each user-end model. Each classifier is parameterized by a user-specific key and distinguishes its associated model distribution from the authentic data distribution. For correct attribution, we expect one-hot classification outcomes for generated contents, and a zero vector for authentic data. To achieve correct attribution, we study the sufficient conditions of the user-specific keys that guarantee an attributability lower bound. The resultant conditions are used to develop an algorithm for computing the keys. Lastly, we assume that attackers can post-process generated contents to potentially deny the attribution, and study the tradeoff between generation quality and robustness of attribution against post-processes.

**Problem formulation**    We assume that for a given dataset $\mathcal{D} \subset \mathbb{R}^{d_x}$, the registry generates user-specific *keys*, $\Phi := \{\phi_1, \phi_2, ...\}$ where $\phi_i \in \mathbb{R}^{d_x}$ and $||\phi_i|| = 1$. $|| \cdot ||$ is the $l_2$ norm. A user-end generative model is denoted by $G_\phi(\cdot; \theta) : \mathbb{R}^{d_z} \to \mathbb{R}^{d_x}$ where $z$ and $x$ are the latent and output variables, respectively, and $\theta$ are the model parameters. When necessary, we will suppress $\theta$ and $\phi$ to reduce the notational burden. The dissemination of the user-end models is accompanied by a public service that tells whether a query content belongs to $G_\phi$ (labeled as 1) or not (labeled as $-1$). We model the underlying binary linear classifier as $f_\phi(x) = \text{sign}(\phi^T x)$. Note that linear models are necessary for the development of sufficient conditions of attribution presented in this paper, although sufficient conditions for nonlinear classifiers are worth exploring in the future.

The following quantities are central to our investigation: (1) *Distinguishability* of $G_\phi$ measures the accuracy of $f_\phi(x)$ at classifying $G_\phi$ against $\mathcal{D}$:

$$D(G_\phi) := \frac{1}{2} \mathbb{E}_{x \sim P_{G_\phi}, x_0 \sim P_\mathcal{D}} \left[ \mathbb{1}(f_\phi(x) = 1) + \mathbb{1}(f_\phi(x_0) = -1) \right]. \tag{1}$$

Here $P_\mathcal{D}$ is the authentic data distribution, and $P_{G_\phi}$ the user-end distribution dependent on $\phi$. $G$ is $(1 - \delta)$-distinguishable for some $\delta \in (0, 1]$ when $D(G) \geq 1 - \delta$. (2) *Attributability* measures the averaged multi-class classification accuracy of each model distribution over the collection $\mathcal{G} := \{G_{\phi_1}, ..., G_{\phi_N}\}$:

$$A(\mathcal{G}) := \frac{1}{N} \sum_{i=1}^{N} \mathbb{E}_{x \sim G_{\phi_i}} \mathbb{1}(\phi_j^T x < 0, \forall\, j \neq i, \phi_i^T x > 0). \tag{2}$$

$\mathcal{G}$ is $(1 - \delta)$-attributable when $A(\mathcal{G}) \geq 1 - \delta$. (3) Lastly, We denote by $G(\cdot; \theta_0)$ (or shortened as $G_0$) the root model trained on $\mathcal{D}$, and assume $P_{G_0} = P_\mathcal{D}$. We will measure the (lack of) *generation quality* of $G_\phi$ by the FID score (Heusel et al., 2017) and the $l_2$ norm of the mean output perturbation:

$$\Delta x(\phi) = \mathbb{E}_{z \sim P_z}[G_\phi(z; \theta) - G(z; \theta_0)], \tag{3}$$

where $P_z$ is the latent distribution.

This paper investigates the following question: *What are the sufficient conditions of keys so that the user-end generative models can achieve distinguishability individually and attributability collectively, while maintaining their generation quality?*

**Contributions**    We claim the following contributions:

1. We develop sufficient conditions of keys for distinguishability and attributability, which connect these metrics with the geometry of the data distribution, the angles between keys, and the generation quality.

2. The sufficient conditions lead to simple design rules for the keys: keys should be (1) data compliant, i.e., $\phi^T x < 0$ for $x \sim P_\mathcal{D}$, and (2) orthogonal to each other. We validate these rules using DCGAN (Radford et al., 2015) and StyleGAN (Karras et al., 2019a) on benchmark datasets including MNIST (LeCun & Cortes, 2010), CelebA (Liu et al., 2015), and FFHQ (Karras et al., 2019a). See Fig. 1 for a visualization of the attributable distributions perturbed from the authentic FFHQ dataset.

3. We empirically test the tradeoff between generation quality and robust attributability under random post-processes including image blurring, cropping, noising, JPEG conversion, and a combination of all.

## 2 SUFFICIENT CONDITIONS FOR ATTRIBUTABILITY

From the definitions (Eq. (1) and Eq. (2)), achieving distinguishability is necessary for attributability. In the following, we first develop the sufficient conditions for distinguishability through Proposition 1 and Theorem 1, and then those for attributability through Theorem 2.

**Distinguishability through watermarking**  First, consider constructing a user-end model $G_\phi$ by simply adding a perturbation $\Delta x$ to the outputs of the root model $G_0$. Assuming that $\phi$ is data-compliant, this model can achieve distinguishability by solving the following problem with respect to $\Delta x$:

$$\min_{||\Delta x|| \leq \varepsilon} \quad \mathbb{E}_{x \sim P_\mathcal{D}} \left[ \max\{1 - \phi^T(x + \Delta x), 0\} \right], \tag{4}$$

where $\varepsilon > 0$ represents a generation quality constraint. The following proposition reveals the connection between distinguishability, data geometry, and generation quality (proof in Appendix A):

**Proposition 1.** Let $d_{max}(\phi) := \max_{x \sim P_\mathcal{D}} |\phi^T x|$. If $\varepsilon \geq 1 + d_{max}(\phi)$, then $\Delta x^* = (1 + d_{max}(\phi))\phi$ solves Eq. (4), and $f_\phi(x + \Delta x^*) > 0, \forall x \sim P_\mathcal{D}$.

**Watermarking through retraining user-end models**  The perturbation $\Delta x^*$ can potentially be reverse engineered and removed when generative models are white-box to users (e.g., when models are downloaded by users). Therefore, we propose to instead retrain the user-end models $G_\phi$ using the perturbed dataset $\mathcal{D}_{\gamma,\phi} := \{G_0(z) + \gamma\phi \mid z \sim P_z\}$ with $\gamma > 0$, so that the perturbation is realized through the model architecture and weights. Specifically, the retraining fine-tunes $G_0$ so that $G_\phi(z)$ matches with $G_0(z) + \gamma\phi$ for $z \sim P_z$. Since this matching will not be perfect, we use the following model to characterize the resultant $G_\phi$:

$$G_\phi(z) = G_0(z) + \gamma\phi + \epsilon, \tag{5}$$

where the error $\epsilon \sim \mathcal{N}(\mu, \Sigma)$. In Sec. 3 we provide statistics of $\mu$ and $\Sigma$ on the benchmark datasets, to show that the retraining captures the perturbations well ($\mu$ close to 0 and small variances in $\Sigma$).

Updating Proposition 1 due to the existence of $\epsilon$ leads to Theorem 1, where we show that $\gamma$ needs to be no smaller than $d_{\max}(\phi)$ in order for $G_\phi$ to achieve distinguishability (proof in Appendix B):

**Theorem 1.** Let $d_{max}(\phi) = \max_{x \in \mathcal{D}} |\phi^T x|$, $\sigma^2(\phi) = \phi^T\Sigma\phi$, $\delta \in [0, 1]$, and $\phi$ be a data-compliant key. $D(G_\phi) \geq 1 - \delta/2$ if

$$\gamma \geq d_{max}(\phi) + \sigma(\phi)\sqrt{\log\left(\frac{1}{\delta^2}\right)} - \phi^T\mu. \tag{6}$$

**Remarks**  The computation of $\sigma(\phi)$ requires $G_\phi$, which in turn requires $\gamma$. Therefore, an iterative search is needed to determine $\gamma$ that is small enough to limit the loss of generation quality, yet large enough for distinguishability (see Alg. 1).

**Attributability**  We can now derive the sufficient conditions for attributability of the generative models from a set of $N$ keys (proof in Appendix C):

**Theorem 2.** Let $d_{min} = \min_{x \in \mathcal{D}} |\phi^T x|$, $d_{max} = \max_{x \in \mathcal{D}} |\phi^T x|$, $\sigma^2(\phi) = \phi^T\Sigma\phi$, $\delta \in [0, 1]$. Let

$$a(\phi, \phi') := -1 + \frac{d_{max}(\phi') + d_{min}(\phi') - 2\phi'^T\mu}{\sigma(\phi')\sqrt{\log\left(\frac{1}{\delta^2}\right)} + d_{max}(\phi') - \phi'^T\mu}, \tag{7}$$

for keys $\phi$ and $\phi'$. Then $A(\mathcal{G}) \geq 1 - N\delta$, if $D(G) \geq 1 - \delta$ for all $G_\phi \in \mathcal{G}$, and

$$\phi^T\phi' \leq a(\phi, \phi') \tag{8}$$

for any pair of data-compliant keys $\phi$ and $\phi'$.

**Remarks**  When $\sigma(\phi')$ is negligible for all $\phi'$ and $\mu = 0$, $a(\phi, \phi')$ is approximately $d_{\min}(\phi')/d_{\max}(\phi') > 0$, in which case $\phi^T\phi' \leq 0$ is sufficient for attributability. In Sec. 3 we empirically show that this approximation is plausible for the benchmark datasets.

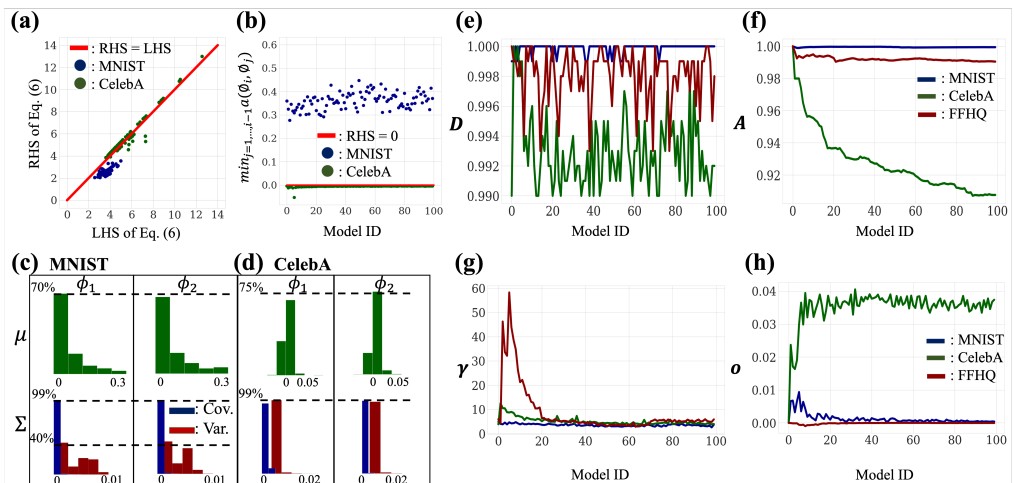

Figure 2: (a) Validation of Theorem 1: All points should be close to the diagonal line or to its right. (b) Support for orthogonal keys: Min. RHS value of Eq.(7) for all keys are either positive (MNIST) or close to zero (CelebA). (c,d) Statistics of $\mu$ and $\Sigma$ for two sample user-end models for MNIST and CelebA. Small $\mu$ and small $diag(\Sigma)$ suggest good match of $G_\phi$ to the perturbed data distributions. (e-h) Distinguishability, attributability, perturbation length, and orthogonality of 100 StyleGAN user-end models on FFHQ and 100 DCGAN user-end models on MNIST or CelebA, respectively.

## 3 EXPERIMENTS AND ANALYSIS

In this section we test Theorem 1, provide empirical support for the orthogonality of keys, and present experimental results on model attribution using MNIST, CelebA, and FFHQ. Note that tests on the theorems require estimation of $\Sigma$, which is costly for models with high-dimensional outputs, and therefore are only performed on MNIST and CelebA.

**Key generation** We generate keys by iteratively solving the following convex problem:

$$\phi_i = \arg\min_\phi \ \mathbb{E}_{x \sim P_D, G_0} \left[ \max\{1 + \phi^T x, 0\} \right] + \sum_{j=1}^{i-1} \max\{\phi_j^T \phi, 0\}. \tag{9}$$

The orthogonality penalty is omitted for the first key. The solutions are normalized to unit $l_2$ norm before being inserted into the next problem. We note that $P_\mathcal{D}$ and $P_{G_0}$ do not perfectly match in practice, and therefore we draw with equal chance from both distributions during the computation. $G_0$s are trained using the standard DCGAN architecture for MNIST and CelebA, and StyleGAN for FFHQ. Training details are deferred to Appendix D.

**User-end generative models** The training of $G_\phi$ follows Alg. 1, where $\gamma$ is iteratively tuned to balance generation quality and distinguishability. For each $\gamma$, we collect a perturbed dataset $\mathcal{D}_{\gamma,\phi}$ and solve the following training problem:

$$\min_\theta \mathbb{E}_{(z,x) \sim \mathcal{D}_{\gamma,\phi}} \left[ ||G_\phi(z; \theta) - x||^2 \right], \tag{10}$$

starting from $\theta = \theta_0$. If the resultant model does not meet the distinguishability requirement due to the discrepancy between $\mathcal{D}_{\gamma,\phi}$ and $G_\phi$, the perturbation is updated as $\gamma = \alpha\gamma$. In experiments, we use a standard normal distribution for $P_z$, and set $\delta = 10^{-2}$ and $\alpha = 1.1$.

**Validation of Theorem 1** Here we validate the sufficient condition for distinguishability. Fig. 2a compares the LHS and RHS values of Eq. (6) for 100 distinguishable user-end models. The empirical distinguishability of these models are reported in Fig. 2e. Calculation of the RHS of Eq. (6) requires estimations of $\mu$ and $\Sigma$. To do this, we sample

$$\epsilon(z) = G_\phi(z; \theta) - G(z; \theta_0) - \gamma\phi \tag{11}$$

using 5000 samples of $z \sim P_z$, where $G_\phi$ and $\gamma$ are derived from Alg. 1. $\Sigma$ and $\mu$ are then estimated for each $\phi$. Fig. 2c and d present histograms of the elements in $\mu$ and $\Sigma$ for two user-end models of the benchmark datasets. Results in Fig. 2a show that the sufficient condition for distinguishability (Eq. (6)) is satisfied for most of the sampled models through the training specified in Alg. 1. Lastly, we notice that the LHS values for MNIST are farther away from the equality line than those for CelebA. This is because the MNIST data distribution resides at corners of the unit box. Therefore perturbations of the distribution are more likely to exceed the bounds for pixel values. Clamping of these invalid pixel values reduces the effective perturbation length. Therefore to achieve distinguishability, Alg. 1 seeks $\gamma$s larger than needed. This issue is less observed in CelebA, where data points are rarely close to the boundaries. Fig. 2g present the values of $\gamma$s of all user-end models.

---

**Algorithm 1:** Training of $G_\phi$

---

**input** : $\phi, G_0$
**output**: $G_\phi, \gamma$
1   set $\gamma = d_{\max}(\phi)$ ;
2   collect $\mathcal{D}_{\gamma,\phi}$ ;
3   train $G_\phi$ by solving Eq. (10) using $\mathcal{D}_{\gamma,\phi}$ ;
4   compute empirical $D(G_\phi)$ ;
5   **if** $D(G_\phi) < 1 - \delta$ **then**
6      set $\gamma = \alpha\gamma$ ;
7      goto step 2 ;
8   **end**

---

**Validation of Theorem 2**   Recall that from Theorem 2, we recognized that orthogonal keys are sufficient. To support this design rule, Fig. 2b presents the minimum RHS values of Eq. (8) for 100 user-end models. Specifically, for each $\phi_i$, we compute $a(\phi_i, \phi_j)$ (Eq. (7)) using $\phi_j$ for $j = 1, ..., i - 1$ and report $\min_j a(\phi_i, \phi_j)$, which sets an upper bound on the angle between $\phi_i$ and all existing $\phi$s. The resultant $\min_j a(\phi_i, \phi_j)$ are all positive for MNIST and close to zero for CelebA. From this result, an angle of $\geq 94 \deg$, instead of $90 \deg$, should be enforced between any pairs of keys for CelebA. However, since the conditions are sufficient, orthogonal keys still empirically achieve high attributability (Fig. 2f), although improvements can be made by further increasing the angle between keys. Also notice that the current computation of keys (Eq. (9)) does not enforce a hard constraint on orthogonality, leading to slightly acute angles ($87.7 \deg$) between keys for CelebA (Fig. 2h). On the other hand, the positive values in Fig. 2b for MNIST suggests that further reducing the angles between keys is acceptable if one needs to increase the total capacity of attributable models. However, doing so would require the derivation of new keys to rely on knowledge about all existing user-end models (in order to compute Eq. (7)).

Table 1: Empirical average of distinguishability ($\bar{D}$), attributability ($A(\mathcal{G})$), $||\Delta x||$, and FID scores. $\text{DCGAN}_M$ ($\text{DCGAN}_C$) for MNIST (CelebA). Std in parenthesis. $\text{FID}_0$: FID for $G_0$. $\downarrow$ means lower is better and $\uparrow$ means higher is better.

| GANs | Angle | $\bar{D} \uparrow$ | $A(\mathcal{G}) \uparrow$ | $||\Delta x|| \downarrow$ | $\text{FID}_0 \downarrow$ | $\text{FID} \downarrow$ |
|---|---|---|---|---|---|---|
| $\text{DCGAN}_M$ | Orthogonal | 0.99 | **0.99** | 3.97 (0.29) | 7.82 (0.15) | 10.43 (0.77) |
| | 45 degree | 0.99 | 0.13 | 3.85 (0.12) | - | 11.02 (0.86) |
| $\text{DCGAN}_C$ | Orthogonal | 0.99 | **0.93** | 4.04 (0.32) | 35.95 (0.12) | 58.69 (5.21) |
| | 45 degree | 0.99 | 0.15 | 4.57 (0.35) | - | 59.81 (5.34) |
| StyleGAN | Orthogonal | 0.99 | **0.99** | 36.04 (23.35) | 12.43(0.16) | 35.23(14.67) |
| | 45 degree | 0.97 | 0.18 | 67.19 (35.53) | - | 47.86(10.66) |

**Empirical results on benchmark datasets**   Tab. 1 reports the metrics of interest measured on the 100 user-end models for each of MNIST and CelebA, and 20 models for FFHQ. All models are trained to be distinguishable. And by utilizing Theorem 2, they also achieve high attributability. As a comparison, we demonstrate results where keys are $45 \deg$ apart ($\phi^T \phi' = 0.71$) using a separate set of 20 user-end models for each of MNIST and CelebA, and 5 models for FFHQ, in which case distinguishability no longer guarantees attributability. Regarding generation quality, $G_\phi$s receive worse FID scores than $G_0$ due to the perturbations. We visualize samples from user-end models and the corresponding keys in Fig. 3. Note that for human faces, FFHQ in particular, the perturbations create light shades around eyes and lips, which is an unexpected but reasonable result.

**Attribution robustness vs. generation quality**   We now consider the scenario where outputs of the generative models are post-processed (e.g., by adversaries) before being attributed. When the post-processes are known, we can take counter measures through robust training, which intuitively

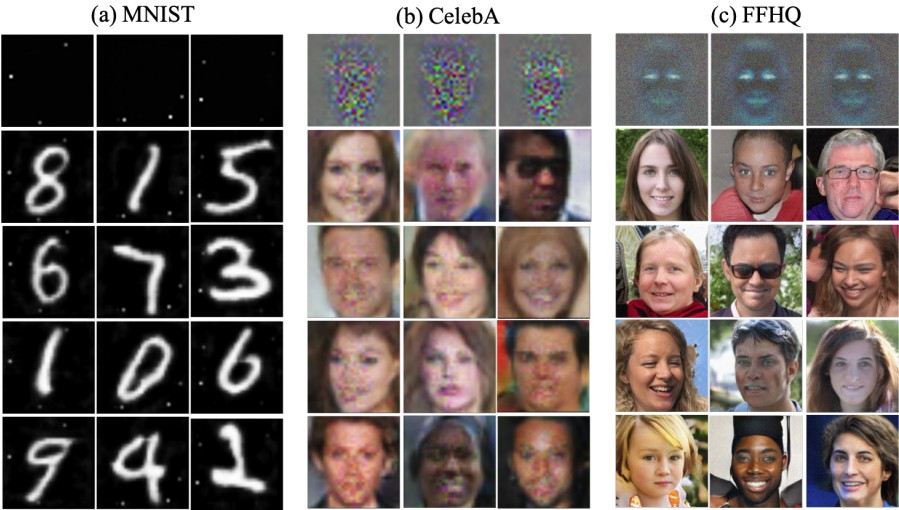

Figure 3: Visualization of sample keys (1st row) and the corresponding user-end generated contents.

will lead to additional loss of generation quality. To assess this tradeoff between robustness and generation quality, we train $G_\phi$ against post-processes $T : \mathbb{R}^{d_x} \to \mathbb{R}^{d_x}$ from a distribution $P_T$. Due to the potential nonlinearity of $T$ and the lack of theoretical guarantee in this scenario, we resort to the following robust training problem for deriving the user-end models:

$$\min_{\theta_i} \mathbb{E}_{z \sim P_z, T \in P_T} \left[ \max\{1 - f_{\phi_i}(T(G_{\phi_i}(z; \theta_i))), 0\} + C||G_0(z) - G_{\phi_i}(z; \theta_i)||^2 \right], \quad (12)$$

where $C$ is the hyper-parameter for generation quality. Detailed analysis and comparison for selecting $C$ are provided in Appendix E. We consider five types of post-processes: blurring, cropping, noise, JPEG conversion and the combination of these four. Examples of the post-processed images are shown in Fig. 5. `Blurring` uses Gaussian kernel widths uniformly drawn from $\frac{1}{3}\{1, 3, 5, 7, 9\}$. `Cropping` crops images with uniformly drawn ratios between $80\%$ and $100\%$, and scales the cropped images back to the original size using bilinear interpolation. `Noise` adds white noise with standard deviation uniformly drawn from $[0, 0.3]$. `JPEG` applies JPEG compression. `Combination` performs each attack with a $50\%$ chance in the order of `Blurring`, `Cropping`, `Noise` and `JPEG`. For differentiability, we use existing implementations of differentiable blurring (Riba et al. (2020)) and JPEG conversion (Zhu et al. (2018)). For robust training, we apply the post-process to mini-batches with $50\%$ probability.

We performed comprehensive tests using DCGAN (on MNIST and CelebA), PGAN (on CelebA), and CycleGAN (on Cityscapes). Tab. 2 summarizes the average distinguishability, the attributability, the perturbation length $||\Delta x||$, and the FID score with and without robust training of $G_\phi$. Results are based on 20 models for each architecture-dataset pair, where keys are kept orthogonal and data compliant. From the results, defense against these post-processes can be achieved, except for `Combination`. Importantly, there is a clear tradeoff between robustness and generation quality. This can be seen from Fig. 5, which compares samples with7 and without robust training from the tested models and datasets.

Lastly, it is worth noting that the training formulation in Eq. (12) can also be applied to the training of non-robust user-end models in place of Eq. (10). However, the resultant model from Eq. (12) cannot be characterized by Eq. (5) with small $\mu$ and $\Sigma$, i.e., due to the nonlinearity of the training process of Eq. (12, the user-end model distribution is deformed while it is perturbed. This resulted in unsuccessful validation of the theorems, which led to the adoption of Eq. (10) for theorem-consistent training. Therefore, while the empirical results show feasibility of achieving robust attributability using Eq. (12, counterparts to Theorems 1 and 2 in this nonlinear setting are yet to be developed.

Table 2: DCGAN$_M$: MNIST. DCGAN$_C$: CelebA. Dis.: Distinguishability before (Bfr) and after (Aft) robust training. Att.: Attributability. $||\Delta x||$ and FID are after robust training. Std in parenthesis. $\downarrow$ means lower is better and $\uparrow$ means higher is better. $||\Delta x||$ and FID before robust training: DCGAN$_M$:$||\Delta x||$ = 5.05, FID = 5.36. DCGAN$_C$:$||\Delta x||$ = 5.63, FID = 53.91. PGAN: $||\Delta x||$ = 9.29, FID = 21.62. CycleGAN: $||\Delta x||$ = 55.85. FID does not apply to CycleGAN.

| Metric | Model | Blurring | | Cropping | | Noise | | JPEG | | Combi. | |
|---|---|---|---|---|---|---|---|---|---|---|---|
| - | - | Bfr | Aft | Bfr | Aft | Bfr | Aft | Bfr | Aft | Bfr | Aft |
| Dis. $\uparrow$ | DCGAN$_M$ | 0.49 | 0.96 | 0.52 | 0.99 | 0.85 | 0.99 | 0.54 | 0.99 | 0.50 | 0.66 |
| | DCGAN$_C$ | 0.49 | 0.99 | 0.49 | 0.99 | 0.95 | 0.98 | 0.51 | 0.99 | 0.50 | 0.85 |
| | PGAN | 0.50 | 0.98 | 0.51 | 0.99 | 0.97 | 0.99 | 0.96 | 0.99 | 0.50 | 0.76 |
| | CycleGAN | 0.49 | 0.92 | 0.49 | 0.87 | 0.98 | 0.99 | 0.55 | 0.99 | 0.49 | 0.67 |
| Att. $\uparrow$ | DCGAN$_M$ | 0.02 | 0.94 | 0.03 | 0.88 | 0.77 | 0.95 | 0.16 | 0.98 | 0.00 | 0.26 |
| | DCGAN$_C$ | 0.00 | 0.98 | 0.00 | 0.99 | 0.89 | 0.93 | 0.07 | 0.98 | 0.00 | 0.70 |
| | PGAN | 0.13 | 0.99 | 0.07 | 0.99 | 0.97 | 0.99 | 0.99 | 0.99 | 0.06 | 0.98 |
| | CycleGAN | 0.08 | 0.98 | 0.05 | 0.93 | 0.97 | 0.98 | 0.47 | 0.99 | 0.05 | 0.73 |
| $||\Delta x|| \downarrow$ | DCGAN$_M$ | 15.96(2.18) | | 9.17(0.65) | | 5.93(0.34) | | 6.48(0.94) | | 17.08(1.86) | |
| | DCGAN$_C$ | 11.83(0.65) | | 9.30(0.31) | | 4.75(0.17) | | 6.01(0.29) | | 13.69(0.59) | |
| | PGAN | 18.49(2.04) | | 21.27(0.81) | | 10.20(0.81) | | 10.08(1.03) | | 24.82(2.33) | |
| | CycleGAN | 68.03(3.62) | | 80.03(3.59) | | 55.47(1.60) | | 57.42(2.00) | | 83.94(4.66) | |
| FID $\downarrow$ | DCGAN$_M$ | 41.11(20.43) | | 21.58(2.44) | | 5.79(0.19) | | 6.50(1.70) | | 68.16(24.67) | |
| | DCGAN$_C$ | 73.62(6.70) | | 98.86(9.51) | | 59.51(1.60) | | 60.35(2.57) | | 87.29(9.29) | |
| | PGAN | 28.15(3.43) | | 47.94(5.71) | | 25.43(2.19) | | 22.86(2.06) | | 45.16(7.87) | |

## 4 DISCUSSION

**Capacity of keys** For real-world applications, we hope to maintain attributability for a large set of keys. Our study so far suggests that the capacity of keys is constrained by the data compliance and orthogonality requirements. While the empirical study showed the feasibility of computing keys through Eq. (9), finding the maximum number of feasible keys is a problem about optimal sphere packing on a segment of the unit sphere (Fig. 4). To explain, the unit sphere represents the identifiability requirement $||\phi|| = 1$. The feasible segment of the unit sphere is determined by the data compliance and generation quality constraints. And the spheres to be packed have radii following the sufficient condition in Theorem 2. Such optimal packing problems are known open challenges (Cohn et al. (2017); Cohn (2016)). For real-world applications where a capacity of attributable models is needed (which is the case for both malicious personation and copyright infringement settings), it is necessary to find approximated solutions to this problem.

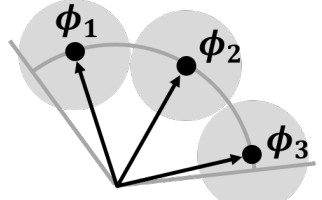

Figure 4: Capacity of keys as a sphere packing problem: The feasible space (arc) is determined by the data compliance and generation quality constraints, and the size of spheres by the minimal angle between keys.

**Generation quality control** From Proposition 1 and Theorem 1, the inevitable loss of generation quality is directly related to the length of perturbation ($\gamma$), which is related to $d_{\max}$. Fig. 6 compares outputs from user-end models with different $d_{\max}$s. While it is possible to filter $\phi$s based on their corresponding $d_{\max}$s for generation quality control, here we discuss a potential direction for prescribing a subspace of $\phi$s within which quality can be controlled. To start, we denote by $J(x)$ the Jacobian of $G_0$ with respect to its generator parameters $\theta_0$. Our discussion is related to the matrix $M = \mathbb{E}_{x \sim P_{G_0}}[J(x)]\mathbb{E}_{x \sim P_{G_0}}[J(x)^T]$. A spectral analysis of $M$ reveals that the eigenvectors of $M$ with large eigenvalues are more structured than those with small ones (Fig. 7(a)). This finding is consistent with the definition of $M$: The largest eigenvectors of $M$ represent the principal axes of all mean sensitivity vectors, where the mean is taken over the latent space. For MNIST, these eigenvectors overlap with the digits; for CelebA, they are structured color patterns. On the other hand, the smallest eigenvectors represent directions rarely covered by the sensitivity vectors, thus

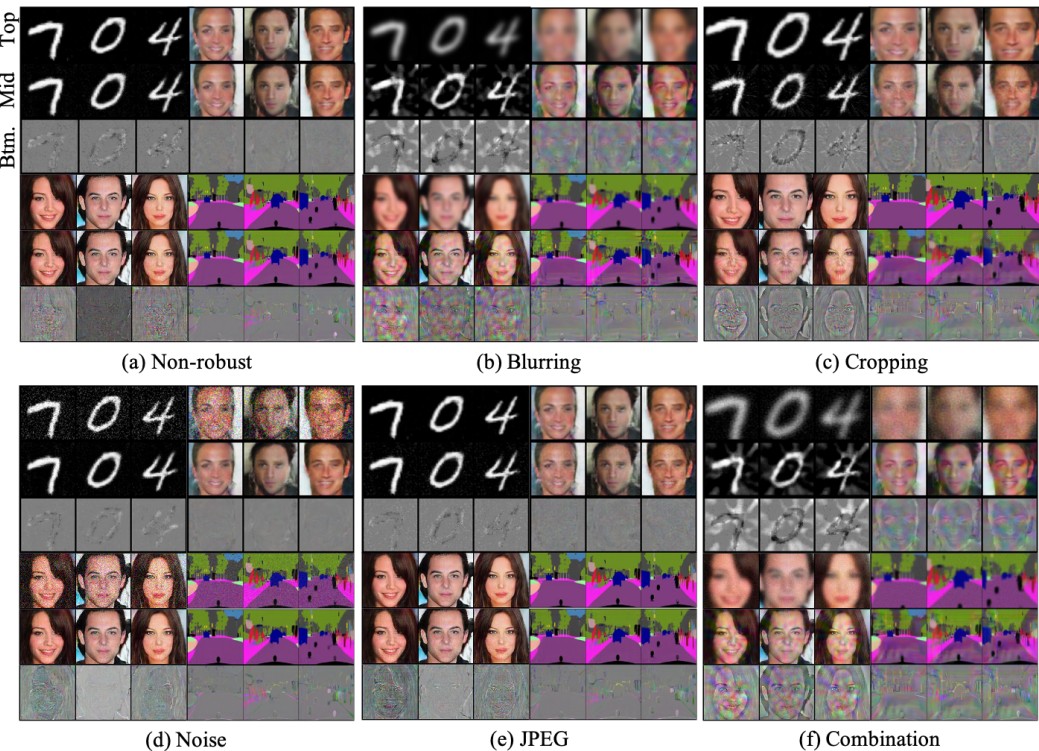

Figure 5: Samples from user-end models with robust and non-robust training. For each subfigure - top: DCGAN on MNIST and CelebA; bottom: PGAN (CelebA) and CycleGAN (Cityscapes). For each dataset - top: samples from $G_0$ (after worst-case post-process in (b-f)); mid: samples from $G_\phi$ (after robust training in (b-f)); btm (a): difference between non-robust $G_\phi$ and $G_0$; btm (b-h) difference between robust and non-robust $G_\phi$.

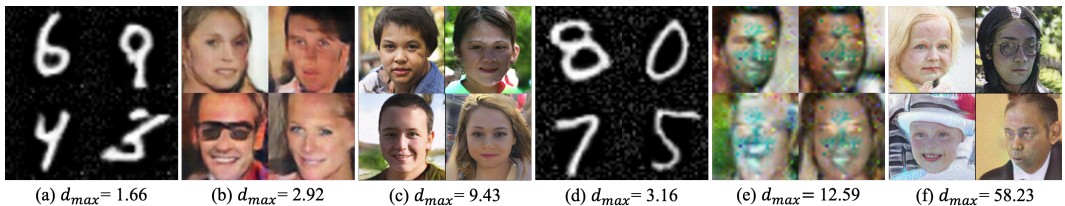

(a) $d_{max}$= 1.66  (b) $d_{max}$= 2.92  (c) $d_{max}$= 9.43  (d) $d_{max}$= 3.16  (e) $d_{max}$= 12.59  (f) $d_{max}$= 58.23

Figure 6: MNIST, CelebA, and FFHQ examples from $G_\phi$s with (a-c) small $d_{max}$ and (d-f) large $d_{max}$. All models are distinguishable and attributable. (Zooming in on pdf file is recommended.)

resembling random noise. Based on this finding, we test the hypothesis that keys more aligned with the eigenspace of the small eigenvalues will have smaller $d_{\max}$. We test this hypothesis by computing the Pearson correlations between $d_{\max}$ and $\phi^T M \phi$ using 100 models for each of MNIST and CelebA. The resultant correlations are 0.33 and 0.53, respectively. In addition, we compare outputs from models using the largest and the smallest eigenvectors of $M$ as the keys in Fig. 7b. While a concrete human study is needed, the visual results suggest that using eigenvectors of $M$ is a promising approach to segmenting the space of keys according to their induced generation quality.

## 5 RELATED WORK

**Detection and attribution of model-generated contents**  This paper focused on the attribution of contents from generative models rather than the detection of hand-crafted manipulations (Agarwal & Farid (2017); Popescu & Farid (2005); O'brien & Farid (2012); Rao & Ni (2016); Huh et al. (2018)). Detection methods rely on fingerprints intrinsic to generative models (Odena et al. (2016); Zhang

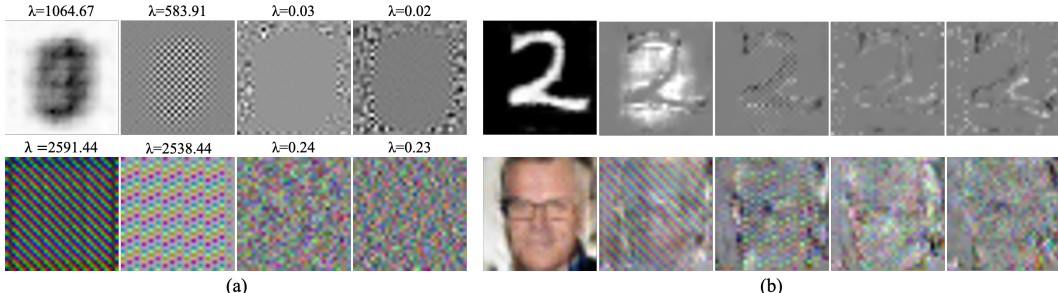

Figure 7: (a) Eigenvectors for the two largest and two smallest eigenvalues of $M$ for DCGANs on MNIST (top) and CelebA (bottom). (b) Left column: Samples from $G_0$; Rest: $G_0 - G_\phi$ where $\phi$ are the eigenvectors in (a).

et al. (2019b); Marra et al. (2019); Wang et al. (2019)), yet similar fingerprints present for models trained on similar datasets (Marra et al. (2019)). Thus fingerprints cannot be used for attribution. Skripniuk et al. (2020) studied decentralized attribution. Instead of using linear classifiers for attribution, they train a watermark encoder-decoder network that embeds (and reads) watermarks into (and from) the content, and compare the decoded watermark with user-specific ones. Their method does not provide sufficient conditions of the watermarks for attributability.

**IP protection of digital contents and models**  Watermarks have conventionally been used for IP protection (Tirkel et al., 1993; Van Schyndel et al., 1994; Bi et al., 2007; Hsieh et al., 2001; Pereira & Pun, 2000; Zhu et al., 2018; Zhang et al., 2019a) without considering the attribution guarantee. Another approach to content IP protection is blockchain (Hasan & Salah, 2019). However, this approach requires meta data to be transferred along with the contents, which may not be realistic in adversarial settings. E.g., one can simply take a picture of a synthetic image to remove any meta data attached to the image file. Aside from the protection of *contents*, mechanisms for protecting IP of *models* have also been studied (Uchida et al., 2017; Nagai et al., 2018; Le Merrer et al., 2019; Adi et al., 2018; Zhang et al., 2018; Fan et al., 2019; Szyller et al., 2019; Zhang et al., 2020). Model watermarking is usually done by adding watermarks into model weights (Uchida et al., 2017; Nagai et al., 2018), by embedding unique input-output mapping into the model (Le Merrer et al., 2019; Adi et al., 2018; Zhang et al., 2018), or by introducing a passport mechanism so that model accuracy drops if the right passport is not inserted (Fan et al., 2019). While closely related, existing work on model IP protection focused on the distinguishability of individual models, rather than the attributability of a model set.

## 6  CONCLUSION

Motivated by emerging challenges with generative models, e.g., deepfake, this paper investigated the feasibility of decentralized attribution of such models. The study is based on a protocol where the registry generates user-specific keys that guides the watermarking of user-end models to be distinguishable from the authentic data. The outputs of user-end models will then be attributed by the registry through the binary classifiers parameterized by the keys. We developed sufficient conditions of the keys so that distinguishable user-end models achieve guaranteed attributability. These conditions led to simple rules for designing the keys. With concerns about adversarial post-processes, we further showed that robust attribution can be achieved using the same design rules, and with additional loss of generation quality. Lastly, we introduced two open challenges towards real-world applications of the proposed attribution scheme: the prescription of the key space with controlled generation quality, and the approximation of the capacity of keys.

## 7  ACKNOWLEDGEMENTS

Support from NSF Robust Intelligence Program (1750082), ONR (N00014-18-1-2761), and Amazon AWS MLRA is gratefully acknowledged. We would like to express our gratitude to Ni Trieu (ASU) for providing us invaluable advice, and Zhe Wang, Joshua Feinglass, Sheng Cheng, Yongbaek Cho and Huiliang Shao for helpful comments.

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

## A  PROOF OF PROPOSITION 1

**Proposition 1.** *Let* $d_{max}(\phi) := \max_{x \sim P_{\mathcal{D}}} |\phi^T x|$. *If* $\varepsilon \geq 1 + d_{max}(\phi)$, *then* $\Delta x = (1 + d_{max}(\phi))\phi$ *solves Eq. (4), and* $f_\phi(x + \Delta x) > 0 \,\forall\, x \sim G_0$.

*Proof.* Let $\phi$ be a data-compliant key and let $x$ be sampled from $P_{\mathcal{D}}$. First, from the KKT conditions for Eq. (4) we can show that the solution $\Delta x^*$ is proportional to $\phi$:

$$\Delta x^* = \phi/\mu^*, \tag{13}$$

where $\mu^* \geq 0$ is the Lagrange multiplier. To minimize the objective, we seek $\mu$ such that

$$1 - (x + \Delta x^*)^T \phi = 1 - x^T \phi - 1/\mu^* \leq 0, \tag{14}$$

for all $x$. Since $x^T \phi < 0$ (data compliance), this requires $1/\mu^* = 1 + d_{max}(\phi)$. Therefore, when $\varepsilon \geq 1 + d_{max}(\phi)$, $\Delta x^* = (1 + d_{max}(\phi))\phi$ solves Eq. (4). And $f_\phi(x + \Delta x^*) = \phi^T(x + (1 + d_{max}(\phi))\phi) = \phi^T x + 1 + d_{max}(\phi) > 0$.

$\square$

## B  PROOF OF THEOREM 1

**Theorem 1.** *Let* $d_{max}(\phi) = \max_{x \in \mathcal{D}} |\phi^T x|$, $\sigma^2(\phi) = \phi^T \Sigma \phi$, $\delta \in [0, 1]$, *and* $\phi$ *be a data-compliant key.* $D(G_\phi) \geq 1 - \delta/2$ *if*

$$\gamma \geq \sigma(\phi)\sqrt{\log\left(\frac{1}{\delta^2}\right)} + d_{max}(\phi) - \phi^T \mu. \tag{15}$$

*Proof.* We first note that due to data compliance of keys, $\mathbb{E}_{x \sim P_{\mathcal{D}}}\left[\mathbb{1}(\phi^T x < 0)\right] = 1$. Therefore $D(G_\phi) \geq 1 - \delta/2$ iff $\mathbb{E}_{x \sim P_{G_\phi}}\left[\mathbb{1}(\phi^T x > 0)\right] \geq 1 - \delta$, i.e., $\Pr(\phi^T x > 0) \geq 1 - \delta_d$ for $x \sim P_{G_\phi}$. We now seek a lower bound for $\Pr(\phi^T x > 0)$. To do so, let $x$ and $x_0$ be sampled from $P_{G_\phi}$ and $P_{G_0}$, respectively. Then we have

$$\begin{aligned}
\phi^T x &= \phi^T(x_0 + \gamma\phi + \epsilon) \\
&= \phi^T x_0 + \gamma + \phi^T \epsilon,
\end{aligned} \tag{16}$$

and

$$\Pr(\phi^T x > 0) = \Pr\left(\phi^T \epsilon > -\phi^T x_0 - \gamma\right). \tag{17}$$

Since $d_{max}(\phi) \geq -\phi^T x_0$, we have

$$\Pr(\phi^T x > 0) \geq \Pr\left(\phi^T \epsilon > d_{max}(\phi) - \gamma\right) = \Pr\left(\phi^T(\epsilon - \mu) \leq \gamma - d_{max}(\phi) + \phi^T \mu\right). \tag{18}$$

The latter sign switching in equation 18 is granted by the symmetry of the distribution of $\phi^T(\epsilon - \mu)$, which follows $\mathcal{N}(0, \phi^T \Sigma \phi)$. A sufficient condition for $\Pr(\phi^T x > 0) \geq 1 - \delta$ is then

$$\Pr\left(\phi^T(\epsilon - \mu) \leq \gamma - d_{max}(\phi) + \phi^T \mu\right) \geq 1 - \delta. \tag{19}$$

Recall the following tail bound of $x \sim \mathcal{N}(0, \sigma^2)$ for $y \geq 0$:

$$\Pr(x \leq \sigma y) \geq 1 - \exp(-y^2/2). \tag{20}$$

Compare equation 20 with equation 19, the sufficient condition becomes

$$\gamma \geq \sigma(\phi)\sqrt{\log\left(\frac{1}{\delta^2}\right)} + d_{max}(\phi) - \phi^T \mu. \tag{21}$$

$\square$

## C  PROOF OF THEOREM 2

**Theorem 2.** *Let* $d_{min} = \min_{x \in \mathcal{D}} |\phi^T x|$, $d_{max} = \max_{x \in \mathcal{D}} |\phi^T x|$, $\sigma^2(\phi) = ||\phi||_{\Sigma}^2$, $\delta \in [0, 1]$. $A(\mathcal{G}) \geq 1 - N\delta$ *if* $D(G) \geq 1 - \delta$ *for all* $G_{\phi} \in \mathcal{G}$ *and for any pair of data-compliant keys* $\phi$ *and* $\phi'$:

$$\phi^T \phi' \leq -1 + \frac{d_{max}(\phi') + d_{min}(\phi') - 2\phi'^T \mu}{\sigma(\phi')\sqrt{\log\left(\frac{1}{\delta^2}\right)} + d_{max}(\phi') - \phi'^T \mu}. \tag{22}$$

*Proof.* Let $\phi$ and $\phi'$ be any pair of keys. Let $x$ and $x_0$ be sampled from $P_{G_{\phi}}$ and $P_{G_0}$, respectively. We first derive the sufficient conditions for $\Pr(\phi'^T x < 0) \geq 1 - \delta$. Since $x = x_0 + \gamma\phi + \epsilon$ for $x \in G_{\phi}$, we have

$$\begin{aligned}
\phi'^T x &= \phi'^T (x_0 + \gamma\phi + \epsilon) \\
&= \phi'^T x_0 + \gamma\phi^T \phi' + \phi'^T \epsilon.
\end{aligned} \tag{23}$$

Then

$$\begin{aligned}
\Pr(\phi'^T x < 0) &= \Pr\left(\phi'^T \epsilon < -\phi'^T x_0 - \gamma\phi^T \phi'\right) \\
&\geq \Pr\left(\phi'^T(\epsilon - \mu) < d_{\min}(\phi') - \gamma\phi^T \phi' - \phi'^T \mu\right),
\end{aligned} \tag{24}$$

where $d_{\min}(\phi') := \min_{x \in \mathcal{D}} |\phi'^T x|$ and $\phi'^T(\epsilon - \mu) \sim \mathcal{N}(0, \sigma^2(\phi'))$. Using the same tail bound of normal distribution and Theorem 1, we have $\Pr(\phi^T x < 0) \geq 1 - \delta$ if

$$\begin{aligned}
&- \gamma\phi^T \phi' \geq \sigma(\phi')\sqrt{\log\left(\frac{1}{\delta^2}\right)} - d_{\min}(\phi') + \phi'^T \mu \\
\Rightarrow \quad &\phi^T \phi' \leq -1 + \frac{d_{\max}(\phi') + d_{\min}(\phi') - 2\phi'^T \mu}{\sigma(\phi')\sqrt{\log\left(\frac{1}{\delta^2}\right)} + d_{\max}(\phi') - \phi'^T \mu}
\end{aligned} \tag{25}$$

Note that $\Pr(A = 1, B = 1) = 1 - \Pr(A = 0) - \Pr(B = 0) + \Pr(A = 0, B = 0) \geq 1 - \Pr(A = 0) - \Pr(B = 0)$ for binary random variables $A$ and $B$. With this, it is straight forward to show that when $\Pr(\phi'^T x < 0) \geq 1 - \delta$ for all $\phi' \neq \phi$, and $\Pr(\phi^T x > 0) \geq 1 - \delta$ for all $\phi$, then $\Pr(\phi^T x > 0, \phi'^T x < 0 \ \forall \phi' \neq \phi) \geq 1 - N\delta$ and $A(\mathcal{G}) \geq 1 - N\delta$.

$\square$

## D  TRAINING DETAILS

### D.1  METHOD

We trained user-end models based on the objective function (Eq.(10) in the main text). For datasets where the root models follow DCGAN and PGAN, the user-end models follow the same architecture. For the FFHQ dataset where StyleGAN is used, we introduce an additional shallow convolutional network as a residual part, which is added to the original StyleGAN output to match with the perturbed datasets $\mathcal{D}_{\gamma,\phi}$. In this case, the training using Eq.(10) is limited to the additional shallow network, while the StyleGAN weights are frozen. More specifically, denoting the combination of convolution, ReLU, and max-pooling by Conv-ReLU-Max, the shallow network consists of three Conv-ReLU-Max blocks and one fully connected layer. All of the convolution layers have 4 x 4 kernels, stride 2, and padding 1. And all of the max-pooling layers have 3 x 3 kernels and stride 2.

### D.2  PARAMETERS

We adopt the Adam optimizer for training. Training hyper-parameters are summarized in Table 3.

Table 3: Hyper-parameters to train keys ($\phi$) and generators ($G_\phi$).

| GANs | Dataset | Batch Size | Learning Rate | $\beta_1$ | $\beta_2$ | Epochs |
|------|---------|-----------|---------------|-----------|-----------|--------|
| DCGAN | MNIST | 16 | 0.001 | 0.9 | 0.99 | 10 |
| DCGAN | CelebA | 64 | 0.001 | 0.9 | 0.99 | 2 |
| StyleGAN | FFHQ | 8 | 0.001 | 0.9 | 0.99 | 5 |

### D.3 TRAINING TIME

All experiments are conducted on V100 Tesla GPUs. Table 4 summarizes the number of GPUs used and the training time for the non-robust models (Eq.(10) in the main text) and robust models (Eq.(12) in the main text). Recall that we chose Eq.(10) for training the non-robust user-end models for consistency with the theorems, although Eq.(12) can be used to achieve attributability in practice, as is shown in the robust attribution study. Therefore, the non-robust training takes longer to due the iteration of $\gamma$ in Alg. 1.

Table 4: Training time (in minute) of one key (Eq.(9) in main text) and one generator (Eq.(10) in main text). DCGAN$_M$: DCGAN for MNIST, DCGAN$_C$: DCGAN for CelebA.

| GANs | GPUs | Key | Non-robust | Blurring | Cropping | Noise | JPEG | Combination |
|------|------|-----|-----------|----------|----------|-------|------|-------------|
| DCGAN$_M$ | 1 | 1.77 | 14 | 4.12 | 3.96 | 4.19 | 5.71 | 5.12 |
| DCGAN$_C$ | 1 | 5.31 | 15 | 10.33 | 9.56 | 10.35 | 10.25 | 10.76 |
| PGAN | 2 | 50.89 | 141.07 | 140.05 | 131.90 | 133.46 | 132.46 | 135.07 |
| CycleGAN | 1 | 20.88 | 16.04 | 16.26 | 15.43 | 15.71 | 15.98 | 16.41 |
| StyleGAN | 1 | 54.23 | 3.12 | - | - | - | - | - |

## E ABLATION STUDY

Here we conduct an ablation study on the hyper-parameter $C$ for the robust training formulation (Eq.(12)). Training with larger $C$ focuses more on generation quality, thus sacrificing distinguishability and attributability. These effects are reported in Table 5 and Table 6. Due to limited time, the results here are averaged over five models for each $C$ and data-model pairs.

Table 5: Distinguishability (top), attributability (btm) before (Bfr) and after (Aft) robust training. DCGAN$_M$: DCGAN for MNIST, DCGAN$_C$: DCGAN for CelebA.

| Model | $C$ | Blurring | | Cropping | | Noise | | JPEG | | Combination | |
| - | - | Bfr | Aft | Bfr | Aft | Bfr | Aft | Bfr | Aft | Bfr | Aft |
| DCGAN$_M$ | 10 | 0.49 | **0.97** | 0.51 | **0.99** | 0.84 | **0.99** | 0.53 | **0.99** | 0.50 | **0.63** |
| DCGAN$_M$ | 100 | 0.49 | 0.61 | 0.51 | 0.98 | 0.76 | 0.98 | 0.53 | 0.99 | 0.50 | 0.52 |
| DCGAN$_M$ | 1K | 0.49 | 0.50 | 0.51 | 0.81 | 0.69 | 0.91 | 0.53 | 0.97 | 0.50 | 0.51 |
| DCGAN$_C$ | 10 | 0.49 | **0.99** | 0.49 | **0.99** | 0.96 | **0.99** | 0.50 | **0.99** | 0.49 | **0.85** |
| DCGAN$_C$ | 100 | 0.50 | 0.96 | 0.49 | 0.99 | 0.92 | 0.93 | 0.50 | 0.99 | 0.49 | 0.61 |
| DCGAN$_C$ | 1K | 0.50 | 0.62 | 0.49 | 0.97 | 0.88 | 0.91 | 0.50 | 0.99 | 0.49 | 0.51 |
| PGAN | 100 | 0.50 | **0.98** | 0.50 | **0.99** | 0.96 | **0.99** | 0.96 | **0.99** | 0.50 | **0.81** |
| PGAN | 1K | 0.50 | 0.89 | 0.49 | 0.95 | 0.94 | 0.95 | 0.88 | 0.99 | 0.50 | 0.60 |
| PGAN | 10K | 0.50 | 0.61 | 0.50 | 0.76 | 0.89 | 0.90 | 0.76 | 0.98 | 0.50 | 0.51 |
| CycleGAN | 1K | 0.49 | **0.92** | 0.50 | **0.87** | 0.98 | **0.99** | 0.55 | **0.99** | 0.49 | **0.62** |
| CycleGAN | 10K | 0.49 | 0.70 | 0.50 | 0.66 | 0.94 | 0.96 | 0.52 | 0.98 | 0.50 | 0.51 |
| DCGAN$_M$ | 10 | 0.02 | **0.94** | 0.03 | **0.88** | 0.77 | **0.95** | 0.16 | **0.98** | 0.00 | **0.26** |
| DCGAN$_M$ | 100 | 0.00 | 0.87 | 0.00 | 0.85 | 0.73 | 0.90 | 0.10 | 0.95 | 0.00 | 0.13 |
| DCGAN$_M$ | 1K | 0.00 | 0.75 | 0.00 | 0.80 | 0.63 | 0.80 | 0.10 | 0.91 | 0.00 | 0.05 |
| DCGAN$_C$ | 10 | 0.00 | **0.98** | 0.00 | **0.99** | 0.89 | **0.93** | 0.07 | **0.98** | 0.00 | **0.70** |
| DCGAN$_C$ | 100 | 0.00 | 0.95 | 0.00 | 0.93 | 0.82 | 0.85 | 0.02 | 0.93 | 0.00 | 0.61 |
| DCGAN$_C$ | 1K | 0.00 | 0.90 | 0.00 | 0.89 | 0.77 | 0.81 | 0.00 | 0.88 | 0.00 | 0.43 |
| PGAN | 100 | 0.26 | **1.00** | 0.21 | **1.00** | 0.99 | **0.99** | 0.99 | **0.99** | 0.00 | **0.99** |
| PGAN | 1K | 0.21 | 0.99 | 0.00 | 0.99 | 0.97 | 0.98 | 0.98 | 0.99 | 0.00 | 0.54 |
| PGAN | 10K | 0.00 | 0.51 | 0.00 | 0.90 | 0.90 | 0.92 | 0.83 | 0.99 | 0.00 | 0.22 |
| CycleGAN | 1K | 0.00 | **0.99** | 0.00 | **0.97** | 0.97 | **0.99** | 0.45 | **0.99** | 0.00 | **0.77** |
| CycleGAN | 10K | 0.00 | 0.87 | 0.00 | 0.77 | 0.95 | 0.96 | 0.30 | 0.99 | 0.00 | 0.31 |

Table 6: $||\Delta x||$ (top) and FID score (btm). Standard deviations in parenthesis. DCGAN$_M$: DCGAN for MNIST, DCGAN$_C$: DCGAN for CelebA, Combi.: Combination attack. *Lower is better.*

| Model | $C$ | Baseline | Blurring | Cropping | Noise | JPEG | Combi. |
| - | - | - | - | - | - | - | - |
| DCGAN$_M$ | 10 | 5.05(0.09) | 15.96(2.18) | 9.17(0.65) | 5.93(0.34) | 6.48(0.94) | 17.08(1.86) |
| DCGAN$_M$ | 100 | 4.09(0.53) | 12.95(4.47) | 7.62(1.55) | 4.57(0.78) | 4.70(1.02) | 12.70(3.37) |
| DCGAN$_M$ | 1K | **3.88(0.60)** | **7.17(2.10)** | **7.43(1.37)** | **4.22(0.77)** | **5.12(1.94)** | **7.56(1.41)** |
| DCGAN$_C$ | 10 | 5.63(0.11) | 11.83(0.65) | 9.30(0.31) | 4.75(0.17) | 6.01(0.29) | 13.69(0.59) |
| DCGAN$_C$ | 100 | 3.08(0.27) | 10.00(1.61) | 7.80(0.58) | 3.20(0.45) | 4.26(0.59) | 11.65(1.48) |
| DCGAN$_C$ | 1K | **2.55(0.36)** | **7.68(1.53)** | **7.13(0.47)** | **2.65(0.24)** | **3.39(0.58)** | **9.23(1.22)** |
| PGAN | 100 | 9.29(0.95) | 18.49(2.04) | 21.27(0.81) | 10.20(0.81) | 10.08(1.03) | 24.82(2.33) |
| PGAN | 1K | 6.52(1.85) | 14.79(4.15) | 18.88(1.96) | 6.40(1.48) | 7.09(1.62) | 22.09(2.12) |
| PGAN | 10K | **5.04(1.63)** | **10.19(2.87)** | **18.23(0.94)** | **5.13(1.14)** | **5.67(1.62)** | **17.26(1.39)** |
| CycleGAN | 1K | 55.85(3.67) | 68.03(3.62) | 80.03(3.59) | 55.47(1.60) | 57.42(2.00) | 83.94(4.66) |
| CycleGAN | 10K | **49.66(5.01)** | **58.64(3.70)** | **66.05(3.47)** | **53.14(0.44)** | **54.52(2.30)** | **66.24(5.29)** |
| DCGAN$_M$ | 10 | 5.36(0.12) | 41.11(20.43) | 21.58(2.44) | 5.79(0.19) | 6.50(1.70) | 68.16(24.67) |
| DCGAN$_M$ | 100 | 5.32(0.11) | 23.83(14.29) | 18.39(3.70) | 5.41(0.18) | 5.46(0.11) | 36.05(16.20) |
| DCGAN$_M$ | 1K | **5.23(0.12)** | **10.85(4.28)** | **18.08(1.77)** | **5.37(0.14)** | **5.30(0.96)** | **21.86(4.16)** |
| DCGAN$_C$ | 10 | 53.91(2.20) | 73.62(6.70) | 98.86(9.51) | 59.51(1.60) | 60.35(2.57) | 87.29(9.29) |
| DCGAN$_C$ | 100 | 45.02(3.37) | 73.12(11.03) | 85.50(12.25) | 47.60(2.57) | 50.48(4.58) | 78.11(12.95) |
| DCGAN$_C$ | 1K | **40.85(3.41)** | **55.63(7.97)** | **72.11(13.81)** | **40.87(3.03)** | **45.46(5.03)** | **57.13(7.20)** |
| PGAN | 100 | 21.62(1.73) | 28.15(3.43) | 47.94(5.71) | 25.43(2.19) | 22.86(2.06) | 45.16(7.87) |
| PGAN | 1K | 19.05(3.14) | 25.19(5.26) | 43.48(12.24) | 19.20(2.96) | 19.05(2.82) | 35.07(8.72) |
| PGAN | 10K | **16.75(1.87)** | **18.96(2.65)** | **37.01(8.74)** | **16.94(1.89)** | **17.39(2.33)** | **26.63(4.44)** |

