# OpenReview forum: " Decentralized Attribution of Generative Models"
_ICLR.cc/2021/Conference — ICLR 2021 Poster_

### Official Review · AnonReviewer3 · 2020-10-28
**Very useful concept, however, only shows results on MNIST/CelebA**

**Rating:** 6
**Confidence:** 2

**Review:**


Given the rebuttal, I am willing to raise my score to a 6 due to the added StyleGAN, PGAN, and other experiments and improved paper layout / clarity. The added experiments are welcome addition to the paper and demonstrate this technique. The lip and eyestaining are interesting results and I do hope this direction gets explored in the future.


Summary
==============================================================

This paper tries to create a decentralized method of attributing user-end generative model. The authors explore the conditions needed for these keys to be attributable and an example on how to design the keys for this condition.

Score:
===============================================================


While this work is clearly worthwhile and useful as more generative models are deployed, the chosen toy examples may be a little too simplistic and low resolution to test this technique. Are these keys still attributable as these datasets continue to grow larger. I would raise more score if the technique was shown to work well on datasets more complex than MNIST and Celeb-A. Would the results holds on higher resolution datasets of greater variety?

I would like to see if the keys work for StyleGAN based models that are a very accessible way of generating fake profile pictures, for example https://apnews.com/article/bc2f19097a4c4fffaa00de6770b8a60d

I am unconvinced that this paper should be accepted. Given more experiments on more relevant / diverse datasets and improvements to figures in the paper, I could be persuaded to change my review.
Pros:
=================================================================
- The paper itself is well written
- The demonstrates the keys can be made robust to common end user augmentations (cropping, compression, etc...)


Cons + Improvements:
================================================================
- The technique is only tested on datasets of CelebA and MNIST . Would like to see results on FFHQ or other ones that are used more often by prospective actors.
- The paper only tests on one architecture DCGAN. I would like to see results on StyleGAN/StyleGANv2 architecture since its much more helpful for those models to attributable due to their widespread use. It also opens up some interesting scientific questions about which latent space the keys should be embedded in or if the multiscale training impedes this attribution regime.
- The low resolution of the datasets
- Figures & tables can be difficult to read

Misc fixes / Questions:
================================================================
- The furthest left column for Table 1 and 2 are useless since you only test on one architecture. Consider a multicell label for readability.
- Table II should have more labels into which values of the table are distinguishability and which are FID.
- Typo: higer Table II
- Unlabeled graph next to Algorithm I has axises that are difficult to read.
- Notations for Figure 2 (c) are difficult to read (too small).
- Are the red lines in Figure 2 (b) suppose to be best fit lines?

---

> ### Author Response · Authors · 2020-11-25
> **Response to R3**
>
> **Q1**: The technique is only tested on datasets of CelebA and MNIST . Would like to see results on FFHQ or other ones that are used more often by prospective actors. The paper only tests on one architecture DCGAN. I would like to see results on StyleGAN/StyleGANv2 architecture since its much more helpful for those models to attributable due to their widespread use. It also opens up some interesting scientific questions about which latent space the keys should be embedded in or if the multiscale training impedes this attribution regime.
>
> **A1**: Thank you for this excellent comment. In the revision, we added results from FFHQ using StyleGAN. We show that in this case, the keys that satisfy the sufficient conditions for attributability form a variety of eye shadows and lip stains. An unexpected but reasonable result! So to the concern of whether multiscale training impedes attribution, it does not for FFHQ. Yet more testing is required for follow-up studies.
>
> We also successfully tested PGAN and CycleGAN for the robust attribution study, in addition to the DCGANs. StyleGAN has not yet been tested on robust attribution due to limited time.
>
> Lastly, we agree that it is definitely interesting to understand which latent space the keys should be embedded. So far on FFHQ, attributability is achieved for 20 keys derived through the proposed method. More keys on the way (see wall-clock costs for computing keys and models in the supplementary). All of the keys share the same structure (eye shadows and lip stains), which may suggest a limit space for attributable keys. We polished the discussion on approximating the capacity of keys through an optimal packing problem, which is an open challenge.
>
> ---
>
> **Q2**: Unlabeled graph next to Algorithm I has axises that are difficult to read. Notations for Figure 2 (c) are difficult to read (too small).
>
> **A2**: We deeply apologize for the inconvenience! The visual presentation is thoroughly improved in the revision.
>
> ---
>
> **Q3**: Are the red lines in Figure 2 (b) suppose to be best fit lines?
>
> **A3**: The red line in Fig.2a is a diagonal line ($x=y$). For Theorem 1 to hold, all data points in 2a should be close to the diagonal line or to its right. The red line in Fig.2b is $y=0$. For orthogonal keys to ensure attributability, all data points in 2b should be close to 0 or positive.

---

### Official Review · AnonReviewer2 · 2020-10-29
**This paper proposes a decentralized attribution to the generative model trained on the same dataset. The goal is to distinguish the user-end generative models, and thus facilitates the IP-protection. The idea is to use orthogonal keys to distinguish the generated samples from authentic data. Furthermore, this paper provided theoretic insights into the proposed method. Experimental results on MNIST and CelebA datasets backup the claims.**

**Rating:** 5
**Confidence:** 4

**Review:**

This paper proposes a decentralized attribution to the generative model trained on the same dataset. The goal is to distinguish the user-end generative models, and thus facilitates the IP-protection. The idea is to use orthogonal keys to distinguish the generated samples from authentic data. Furthermore, this paper provided theoretic insights into the proposed method. Experimental results on MNIST and CelebA datasets backup the claims.

Strengths:
1.	The author tempts to distinguish the generative models trained on the sample dataset. This problem is interesting and inspiring to the researchers in the related field.

Weaknesses:
1.	This paper is not well-organized and many parts are misleading. For example, above Eq.3, the author assumes P_{G_{0}} = P_{D}. Does the author take the samples generated by the root generator as the authentic dataset? However, in Section 2 above Eq. 4, the author claims that the authentic data does not belong to any generator.
2.	In Eq.4, the key-dependent generator is obtained via adding perturbation to the output of the root model. This setting may be troublesome as :1. These generators are not actually trained. This is different from the problem which this paper tempt to solve. 2. No adversarial loss to guarantee the perturbed data being similar to the authentic data. 2. How to distinguish the samples from different generators.
3.	Since Eq.4 is closely related to adversarial attack, the authors are supposed to discuss their connections in the related works.
4.	The name of ‘decentralized attribution’ is misleading. Decentralized models are something like federated learning, where a ‘center’ model grasps information from ‘decentralized models’. However, the presented work is not related to such decentralization.
5.	Typos: regarding the adversarial generative models ->regarding to the adversarial generative models; along the keys->along with the keys.

---

> ### Author Response · Authors · 2020-11-21
> **Response to R2**
>
> **Q1**: Above Eq.3, the author assumes $P_{G_{0}} = P_{D}$. Does the author take the samples generated by the root generator as the authentic dataset? However, in Section 2 above Eq. 4, the author claims that the authentic data does not belong to any generator.
>
> **A1**: We have improved the writing and the presentation of the paper based on reviewers' comments. Thank you!
>
> As a response to the example given by the reviewer: First, in theoretical derivation, we assume that the distribution of the root generator $G_0$ matches with the authentic data distribution. Our intent was to make Proposition 1 concise, so that readers can grasp the key ideas without going through the auxiliaries necessary in reality. It is important to note that, in the experiments, we do acknowledge the difference between the two, and therefore combine the two in computing the keys (see Eq. (8) in the original submission).
>
> Second, the reviewer correctly pointed out the ambiguity in the original sentence in Sec. 2 above Eq. (4): by "any generator" we really mean any **user-end** generator". The sentence should read: the authentic data does not belong to any of the user-end generators, $G_{\phi}$s, but it matches with $G_0$! Thank you again for this critical finding.
>
> ---
>
> **Q2**: In Eq.4, the key-dependent generator is obtained via adding perturbation to the output of the root model. This setting may be troublesome as :1. These generators are not actually trained. This is different from the problem which this paper tempt to solve. 2. No adversarial loss to guarantee the perturbed data being similar to the authentic data. 3. How to distinguish the samples from different generators.
>
> **A2**:
>
> 1. We would like to point out that Eq. (4) (and Proposition 1) is a simplified solution to attribution which we further polished in the very next paragraph on "watermarking user-end models" on the same page.
>
> 2. Thank you for raising this concern. We will make the following discussion explicit in the revision: First of all, we showed through Theorem 2 that bounded distinguishability is needed for bounded attributability. This suggests that any user-end model has to be **different** from the authentic dataset. However, we do take an effort at closing the gap between $G_{\phi}$ and $G_0$ while ensuring attributability: To explain, Theorem 1 (and Proposition 1) provides a key-dependent lower bound for the deviation of a watermarked distribution from the authentic distribution. Theorem 1 leads to Algorithm 1, which develops user-end models that are distinguishable while being as close to the authentic dataset as possible. To examine the difference between $G_0$ and the resultant $G_{\phi}$s, we compared the FID scores in Table 1 in the original submission (and the revision). A visual comparison on CelebA is also provided in Fig. 5(a). More results on FFHQ to come in the revision.
>
> 3. The following will be made explicit in the revision: For a given image, which comes either from one of the user-end models or the authentic dataset, we use linear binary classifiers parameterized by all the keys to obtain a series of classification outputs. Using Theorem 2, our design of keys ensures that with high probability, the classification outputs form a one-hot vector, from where the true source model is identified. If the classification returns all zeros, the image is attributed as authentic. *We hope this answers your question but please let us know, thank you.*
>
> ---
>
> **Q3**: Since Eq.4 is closely related to adversarial attack, the authors are supposed to discuss their connections in the related works.
>
> **A3**: Thank you for your comment. There is indeed a connection to adversarial attack since our problem formulation is by nature a defense problem. We will make this connection clear in the related work in the revision.
>
> ---
>
> **Q4**: The name of ‘decentralized attribution’ is misleading. Decentralized models are something like federated learning, where a ‘center’ model grasps information from ‘decentralized models’. However, the presented work is not related to such decentralization.
>
> **A4**: Thank you for this important comment! We take the wording of "decentralized attribution" quite literally without considering its ambiguity due to its usage in a cybersecurity context. To reiterate, in this paper, the final attribution decision is made by grasping individual binary decisions from "decentralized" classifiers (parameterized by keys) rather than one centralized multi-class classifier.
>
> We would strongly appreciate it if the reviewer can suggest a less ambiguous title.

---

### Official Review · AnonReviewer1 · 2020-11-01
**interesting theoretic result on model attribution, but applicability might be limited**

**Rating:** 6
**Confidence:** 3

**Review:**

Summary

Fake content produced by generative models is of great concerns. This paper investigates attribution techniques to identify models that generated the content. The key theoretic result is the derivation of the sufficient conditions for decentralized attribution and the design of keys following these conditions. Thee paper shows that decentralized attribution can be achieved when keys are orthogonal to each other, and belonging to a subspace determined by the data distribution. Results are validated on two datasets, MNIST and CelebA.

Strengths

The paper derives the sufficient conditions for decentralized attribution of models generated the content. The idea is to add a uniform and bounded perturbation. The theory is on the data geometry-dependent threshold. There is a tradeoff between distinguishability and generation quality.

Weaknesses

The decentralized attribution problem is not clearly motivated. What are the use cases? Why centralized attribution does not work for these use cases?

The attack model is not clear from the paper. Specifically, who is the attacker and what is the capability? It seems that the attacker has access to the original generative model. It is not clear why the user-end models need to be retrained. Is this trained by the attacker?

The application settings are not clear.  There are many questions on the motivating example "A company develops a GAN model for image post-processing. A third-party organization (the registry) assigns keys to the company, who is then required to embed a watermark to the GAN models according to the keys for users to download. With the keys, the registry can trace the GAN generated images back to the user-end models." What kind of postprocessing? Are these identify preserving changes or generating new faces? Why is there a need for the watermark to depend on user keys?

In robust training, the paper discusses five types of post-processes: blurring, cropping, noise, JPEG conversion and the combination of these four. Are these necessary constraints on what the GAN models can do in order for the attribution techniques to work?

The derivation is on the sufficient conditions. What is the necessary conditions?

Minor comments

Figure 1 is not referenced anywhere in the paper.

Figure 2 caption has no information on what the two rows are. I suppose one is on MNIST and the other is on CelebA.

Decision

The paper seems to present interesting results on decentralized attribution of generative models. However, the paper poorly motivates the problem, use cases, capabilities and limitations of the approach. I can not recommend accept at this point. It is very possible I do not have the context to appreciate the paper. I hope the rebuttal process can make things clear.


=====POST-REBUTTAL COMMENTS========

I applaud the authors for the very detailed response and the efforts in the updated draft. Many of my questions were clarified.  However, there are still important questions remaining.

1. In addressing my Q4, "We do believe that a separate discussion is needed on whether the registry or the user-end devices should retrain the models, given that users can potentially be the attackers in the malicious personation case. This discussion will entail an exploration of the tradeoff between training efficiency and training security of generative models, which is beyond the scope of this paper." This raises the feasibility question of decentralized approach proposed in this paper.

2. In addressing my Q6, "We note that just like problem settings for adversarial attacks, there is a potential mismatch between our expectation of the attackers' capability and their actual capability. Therefore, we cannot assess whether these constraints are necessary, but we acknowledge that they are not sufficient." This needs more investigation.

3. Additionally, in addressing R3, "Lastly, we agree that it is definitely interesting to understand which latent space the keys should be embedded. So far on FFHQ, attributability is achieved for 20 keys derived through the proposed method. More keys on the way (see wall-clock costs for computing keys and models in the supplementary). All of the keys share the same structure (eye shadows and lip stains), which may suggest a limit space for attributable keys. We polished the discussion on approximating the capacity of keys through an optimal packing problem, which is an open challenge."
If the number of keys are limited, this seriously impacts the contribution of this paper.

Overall, I am more positive. I am willing to raise my score to 6. However, the paper is still somewhat borderline.

---

> ### Author Response · Authors · 2020-11-21
> **Response to R1 (Part 1)**
>
> **Q1**: The decentralized attribution problem is not clearly motivated. What are the use cases?
>
> **A1**: Thank you for the comment. The following clarification on the motivation is being added to the review:
>
> The study is motivated by two types of emerging threat models related to the growing applications of generative models, namely, malicious personation (e.g., deepfake) and digital copyright infringement. In the former, the attacker is the person who uses generative models to create and disseminate inappropriate or illegal contents. Attribution allows law enforcers to identify the specific source model (and its owner) of these contents. In the latter, the attacker is the person who takes a copyrighted digital content (e.g., an art piece created through the assistance of a generative model), makes modifications to it, and claims it as their own. In this case, attribution allows the content owner to protect their IP. In both cases, we assume that the attacker can make changes to the generated contents to potentially deny the attribution. Therefore, it is desirable for the attribution to be robust against such changes. Further, we assume that generative models are white-box to the attacker (e.g., through reverse engineering of the distributed software that contains the generative model). Therefore explicit adding watermarks to generated contents can be removed by the attacker.
>
> The above summarizes our motivation to investigate (1) the attribution of generative models, (2) the robustness of attribution, and (3) implicit watermarking through retraining generative models.
>
> We should also reiterate that traditional detection methods (Rossler et al., 2019; Wang et al., 2019) are not suitable for the aforementioned threat models. This is because detection techniques only discriminate between authentic and generated contents, but cannot be used to identify the exact models that generated the contents. This point is echoed by recent studies that share the same attribution problem settings as ours (Yu et al., 2018; Albright et al.,2019; Zhang et al., 2020).
>
> ---
>
> **Q2**: Why centralized attribution does not work for these use cases?
>
> **A2**: Centralized attribution has been investigated under the premise that a *finite* and *fixed* set of generative models is known by the registry  (Yu et al., 2018; Albright et al., 2019; Zhanget al., 2020). However, existing solutions along this line has two intrinsic limitations for real-world applications:
>
> First, without careful design of the user-end generative models (e.g., model retraining through user-specific keys as in this paper), one cannot derive the sufficient conditions for attributability bounds. Having such sufficient conditions is *critical* for forensics and IP protection, due to the potentially high costs/risks associated with false positives and negatives.
>
> Second, it is unreasonable to assume that the registry has access to the *entire* set of generative models, the size of which can potentially grow arbitrarily. There is also a concern about the computational cost as the multi-class classifier used for attribution will need to be trained on all data generated on all models. In comparison, our method only requires the registry to have access to all user-specific keys for attribution and for generating new keys. Key-specific generative models, which can be trained on the user side, will be intrinsically attributable when the corresponding keys satisfy the sufficient conditions developed in the paper.
>
> ---
>
> References:
>
> Andreas Rossler, Davide Cozzolino, Luisa Verdoliva, Christian Riess, Justus Thies, and Matthias Nießner.  Face forensics++:  Learning to detect manipulated facial images.  In Proceedings of the IEEE International Conference on Computer Vision, pp. 1–11, 2019.
>
> Sheng-Yu Wang, Oliver Wang, Richard Zhang, Andrew Owens, and Alexei A Efros. CNN-generated images are surprisingly easy to spot... for now. arXiv preprint arXiv:1912.11035, 2019.
>
> Baiwu Zhang, Jin Peng Zhou, Ilia Shumailov, and Nicolas Papernot.  Not my deepfake:  Towards plausible deniability for machine-generated media.arXiv preprint arXiv:2008.09194, 2020.
>
> Ning Yu, Larry Davis, and Mario Fritz.  Attributing fake images to GANs: Analyzing fingerprints ingenerated images. arXiv preprint arXiv:1811.08180, 2018.
>
> Michael Albright, Scott McCloskey, and ACST Honeywell.  Source generator attribution via inversion. arXiv preprint arXiv:1905.02259, 2019.

---

> ### Author Response · Authors · 2020-11-21
> **Response to R1 (Part 2)**
>
> **Q3**: The attack model is not clear from the paper. Specifically, who is the attacker and what is the capability? It seems that the attacker has access to the original generative model.
>
> **A3**: Thank you again for the question. We described the threat models and attacker capabilities in Q1. We assume that the attacker has white-box access to the user-end models, the outputs of which are watermarked, but does not have access to the original (root) generative model or the dataset used for training the models.
>
> ---
>
> **Q4**: It is not clear why the user-end models need to be retrained. Is this trained by the attacker?
>
> **A4**: Thank you for your question. The output distribution of the user-end model is supposed to be perturbed from that of the root model ($G_0$), where the perturbation is prescribed by the key ($\phi$). Specifically, the user-end model $G_{\phi}$ should match with {$x+ \gamma \phi | x \sim G_0$}, where $\gamma$ is chosen so that $G_{\phi}$ is distinguishable. The reason why we retrain weights of $G_{\phi}$ (from $G_0$), rather than simply treat $x + \gamma \phi | x\sim G_0$ as a user-end model is because we assume the user-end model to be white-box to the attacker, in which case the attacker can deny the attribution by removing the watermark ($\gamma \phi$).
>
> To the question of who performs the retraining: the retraining can be done by either the registry, who creates and holds the keys, or by the users, who are assigned the keys. We do believe that a separate discussion is needed on whether the registry or the user-end devices should retrain the models, given that users can potentially be the attackers in the malicious personation case. This discussion will entail an exploration of the tradeoff between training efficiency and training security of generative models, which is beyond the scope of this paper.
>
> ---
>
> **Q5**: The application settings are not clear. There are many questions on the motivating example "A company develops a GAN model for image post-processing. A third-party organization (the registry) assigns keys to the company, who is then required to embed a watermark to the GAN models according to the keys for users to download. With the keys, the registry can trace the GAN generated images back to the user-end models." What kind of postprocessing? Are these identify preserving changes or generating new faces? Why is there a need for the watermark to depend on user keys?
>
> **A5**: Thank you for the question. We have refined the motivation in the revision following answers to Q1 and Q2, but would like to address the questions more specifically here.
>
> (1) "Post-processing" is broadly referred to as content generation through generative models (e.g., super-resolution refinement of a photo or conditioned generation of faces). While the experiments in the paper focused on the generation of purely artificial contents (digits and faces), the same attribution technique can be applied to other use cases, e.g., identity preserving post-processing. Mathematically, our method simply perturbs a target distribution in different ways to that the resultant distributions are attributable.
>
> (2) Regarding the dependency of watermarks on keys: In our method, watermarks are vectors proportional to the user keys, and the attribution of watermarked contents is based on a collection of linear classifiers parameterized by the keys. This dependency of watermarks on keys allows us to develop sufficient conditions about keys that guarantee attributability of the watermarked contents.
>
> ---
>
> **Q6**: In robust training, the paper discusses five types of post-processes: blurring, cropping, noise, JPEG conversion and the combination of these four. Are these necessary constraints on what the GAN models can do in order for the attribution techniques to work?
>
> **A6**: We consider that the attacker can perform these post-processes as an attempt to evade attribution. Therefore, we assume that the attribution is only successful when it is robust against these post-processes. This setting matches with the `"`relaxed attribution" definition in (Zhang et al., 2020). We note that just like problem settings for adversarial attacks, there is a potential mismatch between our expectation of the attackers' capability and their actual capability. Therefore, we cannot assess whether these constraints are necessary, but we acknowledge that they are not sufficient.
>
> ---
>
> **Q7**: The derivation is on the sufficient conditions. What is the necessary conditions?
> **A7**: This  is  an  interesting  question  and  worth  exploring.   However,  to  obtain  attributability certificates, what we need are the sufficient conditions.

---

### Author Response · Authors · 2020-11-25
**Short summary of changes**

To all reviewers: Thank you again for your comments and suggestions. We have now submitted the rebuttal revision and supplementary materials, with major changes __highlighted__ in response to your comments. Here is a short summary of these changes:

1. Motivating real-world applications are explained.
2. Results on FFHQ+StyleGAN are added to further support the proposed method.
3. Writing and presentation thoroughly improved.

---

### Decision · Program_Chairs · 2021-01-07
**Final Decision**

**Decision:**

Accept (Poster)

**Comment:**

This paper proposes a decentralized attribution method to distinguish the generative models trained on the sample dataset. The key theoretic result is the derivation of the sufficient conditions for decentralized attribution and the design of keys following these conditions. Results are validated on two datasets with several generative models. The work is very interesting.
R1: Overall, I am more positive. I am willing to raise my score to 6. However, the paper is still somewhat borderline.
R3: Given the rebuttal, I am willing to raise my score to a 6 due to the added StyleGAN, PGAN, and other experiments and improved paper layout / clarity. The added experiments are welcome addition to the paper and demonstrate this technique. The lip and eyestaining are interesting results and I do hope this direction gets explored in the future.